# Reactivity of Coordinated 2-Pyridyl Oximes: Synthesis, Structure, Spectroscopic Characterization and Theoretical Studies of Dichlorodi{(2-Pyridyl)Furoxan}Zinc(II) Obtained from the Reaction between Zinc(II) Nitrate and Pyridine-2-Chloroxime

**Sokratis T. Tsantis** [1] , **Vlasoula Bekiari** [2] , **Demetrios I. Tzimopoulos** [3], **Catherine P. Raptopoulou** [4] , **Vassilis Psycharis** [4,*] , **Athanasios Tsipis** [5,*] **and Spyros P. Perlepes** [1,6,*]

1   Department of Chemistry, University of Patras, 26504 Patras, Greece; sokratis.t.tsantis@gmail.com
2   Department of Crop Science, University of Patras, 30200 Messolonghi, Greece; bbekiari@upatras.gr
3   Department of Chemistry, Aristotle University of Thessaloniki, 54124 Thessaloniki, Greece; dimtzim@auth.gr
4   Institute of Nanoscience and Nanotechnology, NCSR "Demokritos", 15310 Aghia Paraskevi Attikis, Greece; c.raptopoulou@inn.demokritos.gr
5   Department of Chemistry, University of Ioannina, 45110 Ioannina, Greece
6   Institute of Chemical Engineering Sciences, Foundation for Research and Technology-Hellas (FORTH/ICE-HT), Platani, P.O. Box 1414, 26504 Patras, Greece
*   Correspondence: v.psycharis@inn.demokritos.gr (V.P.); attsipis@uoi.gr (A.T.); perlepes@patreas.upatras.gr (S.P.P.); Tel.: +30-210-6503346 (V.P.); +30-26510-08333 (A.T.); +30-2610-996730 (S.P.P.)

**Abstract:** This work reports our first results in the area of the reactivity of coordinated chloroximes. The 1:2:2:2 $Zn(NO_3)_2 \cdot 6H_2O/Eu(NO_3)_3 \cdot 6H_2O/ClpaoH/Et_3N$ reaction mixture in MeOH, where ClpaoH is pyridine-2-chloroxime, resulted in complex $[ZnCl_2(L)]$ (**1**); L is the di(2-pyridyl)furoxan [3,4-di(2-pyridyl)-1,2,5-oxadiazole-2-oxide] ligand. The same complex can be isolated in the absence of the lanthanoid. The direct reaction of $ZnCl_2$ and pre-synthesized L in MeOH also provides access to **1**. In the tetrahedral complex, L behaves as a $N_{pyridyl},N'_{pyridyl}$-bidentate ligand, forming an unusual seven-membered chelating ring. The Hirshfeld Surface analysis of the crystal structure reveals a multitude of intermolecular interactions, which generate an interesting 3D architecture. The complex has been characterized by FTIR and Raman spectroscopies. The structure of **1** is not retained in DMSO (dimethylsulfoxide) solution, as proven by NMR ($^1H$, $^{13}C$, $^{15}N$) spectroscopy and its molar conductivity value. Upon excitation at 375 nm, solid **1** emits blue light with a maximum at 452 nm; the emission is of an intraligand character. The geometric and energetic profiles of possible pathways involved in the reaction of ClpaoH and $Zn(NO_3)_2 \cdot 6H_2O$ in MeOH in the presence of $Et_3N$ has been investigated by DFT (Density Functional Theory) computational methodologies at the PBE0/Def2-TZVP(Cr)∪6-31G(d,p)(E)/Polarizable Continuum Model (PCM) level of theory. This study reveals an unprecedented cross-coupling reaction between two coordinated 2-pyridyl nitrile oxide ligands.

**Keywords:** cross-coupling reaction; DFT mechanistic studies; di(2-pyridyl)furoxan metal complexes; Hirshfeld surface analysis; pyridine-2-chloroxime; reactivity of coordinated oximes

## 1. Introduction

The ability of metal ions to alter the reactivity of ligands toward external reagents is at the heart of their role as catalytic centers in chemistry and biology. Thus, the reactivity of coordinated ligands is a central research theme in modern inorganic and bioinorganic chemistry [1,2]. The reactivity of the coordinated oxime group has been attracting the intense interest of several inorganic chemistry groups around the world [3–8], including our group [9].

Oximes are weak organic acids that have been known about for a long time. Because of their geometrical flexibility and variable electronic structure, organic oximes are used in a variety of reactions to generate a plethora of new organic compounds with useful properties [7–16].

The fantastic introduction of oximes into metal chemistry started with the use of 1,2-napthoquinone monoxime for the quantitative determination of metal ions [17], followed by the famous gravimetric determination of Ni(II) as the red bis(dimethylglyoximato)nickel(II) precipitate [18]. Oxime and oximato metal complexes have played remarkable roles in several aspects of coordination chemistry [19–21], bioinorganic chemistry [22–25], molecular magnetism [26–28] and catalysis [29,30]. The reactivity of coordinated oximes (in other words, metal ion-induced/promoted/assisted/mediated reactions of oximes) present a special interest [3–7]. A coordinated oxime group has three potentially reactive sites (C, N and O atoms) and an –OH group, the acidity of which increases significantly upon coordination [3]. Virtually all types of metal-mediated transformations (nucleophilic and electrophilic reactions, intramolecular redox reactions, template synthesis, Beckman rearrangement, etc.) are possible for the coordinated oximes [6,7]. Metal-mediated reactions of the oxime group involve its O-functionalization, N-functionalization and C-functionalization, dehydration of aldoximes (formation of nitriles and Beckman rearrangement leading to carboxamides), reduction of oxime ethers and esters (with or without preservation of the N-O bond) and oxidation of oximes leading to carbonyl compounds [7]. Moreover, metal ion-involving reactions directed to oxime side chains and the metal-involving reactions of oximes that lead to carbo- and heterocycles are important. As far as the generation of heterocycles is concerned, which is the subject of the present work, three-, four-, five-, six- and seven-membered rings can be formed [7].

A central research theme in our group is the study of the coordination chemistry of 2-pyridyl oximes (Figure 1; R = is a non-donor or a donor group), with an emphasis on (a) the magnetic and optical properties of the resulting homo- and heterometallic dinuclear and polynuclear (coordination clusters) complexes [31–33], and (b) the metal-mediated reactivity of these ligands [34–39]. We have very recently extended our work to pyridine-2-chloroxime or 2-pyridyl chloroxime (ClpaoH; R = Cl in Figure 1), the general goal being the study of the C-functionalization of coordinated 2-pyridyl oximes. There are no literature reports of any metal complexes of ClpaoH or Clpao⁻. In metal-free organic chemistry, the success of oxime C-functionalization of oximes is mainly determined by the leaving ability of the R group. However, in practice, only chloroximes (R′C(Cl)=NOH), which are typically synthesized by the chlorination of aldoximes via the Piloty reaction [7,40], are conventional starting materials for the C-functionalization of the oxime moiety. Generally, the metal-mediated C-functionalization of chloroximes has been poorly investigated [7,41,42] because these reactions proceed rather easily even under metal ion-free conditions. In this report, we describe a synthetic investigation of the Ln(III)/Zn(II)/ClpaoH and Zn(II)/ClpaoH reaction systems (Ln = lanthanoid) which both gave a single interesting Zn(II) complex, featuring the (2-pyridly)furoxan [3,4-di(2-pyridyl)-1,2,5-oxadiazole] ligand derived from ClpaoH. The structure of the complex has been unambiguously determined by single-crystal X-ray crystallography and its characterization has been carried out by several spectroscopic techniques; the mechanism of the transformation was also discussed using advanced theoretical methods.

R=H; paoH

R=Me; mepaoH

R=Ph; phpaoH

R=NH₂; (NH₂)paoH

R=CN; (CN)paoH

R= ; dpkoxH

R=Cl; ClpaoH

**Figure 1.** Structural formulae of some 2-pyridyl oxime-type ligands.

## 2. Results and Discussion

### 2.1. Synthetic Comments

Pyridine-2-chloroxime (ClpaoH; R = Cl in Figure 1) was synthesized by modification of the literature methods [43,44] starting from pyridine-2-amidoxime ((NH₂)paoH; R = NH₂ in Figure 1), see Scheme 1. The modified reaction ensures high purity and very good yields. The identity and purity of the compound was confirmed by microanalyses (C, H, N) and spectroscopic (IR, $^1$H and $^{13}$C NMR) methods (vide infra).

**Scheme 1.** The synthesis of pyridine-2-chloroxime (ClpaoH).

We initially came across the Zn(II) complex described in the present work by trying to prepare heterometallic Zn(II)/Ln(III)/Clpao⁻ complexes; complexes containing both Zn$^{II}$ and Ln$^{III}$ centers often exhibit interesting optical and/or magnetic properties. One of our favorable routes for the synthesis of such compounds is a "one-pot" procedure involving a mixture of Zn(II) and 4f-metal starting materials and an organic ligand possessing distinct functionalities (or "pockets") for preferential bonding of the two different metal ions. The various anionic 2-pyridyl oximes are ideal ligands for this synthetic goal, because the hard (HSAB), deprotonated O atom will favor binding to strongly oxophilic Ln$^{III}$ ions, whereas the softer 2-pyridyl and oximate N atoms will favor coordination to the Zn$^{II}$ center. Using, for example, methyl 2-pyridyl ketoxime (mepaoH; R = Me in Figure 1), we had isolated two families of dinuclear complexes, mainly [ZnLn(mepao)₃(mepaoH)₃] (ClO₄)₂ and [ZnLn(NO₃)₂(mepao)₃(mepaoH)] [31], displaying interesting photoluminescence and magnetic properties. We thought that ClpaoH, if remaining intact during the reaction, could behave in a similar manner, giving similar or different heterometallic species. The reaction between Zn(NO₃)₂·6H₂O, Eu(NO₃)₂·6H₂O, ClpaoH and Et₃N (1:2:2:2) in MeOH gave a polycrystalline powder whose IR, Raman and NMR ($^1$H, $^{13}$C) were not consistent with the involvement of ClpaoH or Clpao⁻ in the product. Furthermore, the decomposition of the powder in 2N HNO₃ and subsequent qualitative analysis showed the presence of chlorides (Cl⁻). The quantitative analysis of Zn$^{II}$ in a sample of the product (EDTA, Eriochrome Black T, pH = 10) gave a rather high percentage of ~18%. The crystallization of the complex revealed the formula [ZnCl₂(L)] (**1**) (vide infra), where L is the ligand di(2-pyridyl)furoxan (Figure 2, left).

**Figure 2.** The ligand di(2-pyridyl)furoxan that is present in complex **1** and the general structural formula of furoxans(derivatives of 1,2,5-oxadiazole-2-oxide).

Furoxan (1,2,5-oxadiazole-2-oxide) and its derivatives (Figure 2, right) are important heterocycles, mainly because of their potential NO donor activity, with wide applications in biological, medicinal, materials and synthetic chemistry [45–51]. Concerning the first two areas, furoxans have been shown to exert a variety of NO-related bioactivities, including cytotoxicity, mutagenicity, immunosuppression, central muscle relaxant properties, anticonvulsive effects, monoamine oxidase inhibition, and direct vasodilator and blood pressure-lowering activities [46,47,50]. Furoxans are also superior candidates for energetic materials and are used to construct energetic Metal–Organic Frameworks (MOFs) due to their positive heat of formation and good oxygen balance [48]. They also exhibit rich reactivity including ring–chain tautomerism, thermal ring cleavage, reactions with nucleophiles and reducing agents, heterocyclic ring transformations and reactions of the substituent attached to ring carbon atoms [49,51]. The furoxan ring can be constructed by various methods, the most synthetically useful of which are the oxidative cyclization of 1,2-dioximes, the dehydration of α-nitroketoximes and, for symmetrically substituted furoxans (R = R′ in Figure 2), the dimerization of nitrile oxides [49,51]. The coordination chemistry of furoxans has been poorly investigated [47,48,52–55].

Having the crystal structure of **1** at hand, we were glad to realize that only one complex of L had been structurally characterized. This is compound [Cu$_2$Cl$_4$(L)$_2$] (**2**) [52], which was synthesized by the 1:1 reaction between CuCl$_2$·2H$_2$O and the pre-synthesized (as described in [43]) L. Since **1** does not contain Eu(III), we repeated the reaction that leads to the complex, but without addition of Eu(NO$_3$)$_3$·6H$_2$O this time. Again, compound **1** was isolated (evidences from analytical data, IR, $^1$H NMR spectroscopies and proof from cell determination) in a comparable yield (~60%). This means that Eu(III) is not essential for the observed ClpaoH→L transformation. To answer the question as to whether the reaction that leads to **1** is Zn$^{II}$-mediated, we performed many "blind" experiments, i.e., crystallizations from 1:1 ClpaoH/Et$_3$N solutions in MeOH under various conditions (the storage of concentrated solutions at low temperatures, solvent evaporation at room temperature, liquid or vapor diffusion of Et$_2$O into the methanolic solution). The determination of the melting points of the obtained solids (the pure L has a melting point of 169–170 °C [43]) and spectroscopic data suggested that none of the solids represented pure L. We could not crystallize the solids (which might be mixtures of products) to obtain structural proof of their identity. These experimental observations do not necessarily mean that the observed transformation is Zn$^{II}$-assisted. As would be expected, the complex can be prepared by a 1:1 reaction between ZnCl$_2$ and pre-synthesized L [43] in MeOH. To further elucidate the mechanism of the interesting reaction that gives **1**, we carried out a detailed theoretical study (vide infra).

During the writing of the present paper, the structure of another Cu(II) complex of L was reported [53]. This is compound [Cu(NO$_3$)$_2$(L)] (**3**), which was prepared form the reaction of Cu(NO$_3$)$_2$·3H$_2$O and the dioxime ligand (1$E$,2$E$)-1,2-di(pyridine-2yl)ethane-1,2-dione dioxime (L′H$_2$) in MeOH under aerobic conditions (Scheme 2). The furoxan ring of L is generated during the formation of the complex from the in situ intramolecular cyclization of the dioxime moiety of the initial ligand L′H$_2$. Through DFT studies, it was proposed that L′H$_2$ is not an acidic molecule, but its acidity is enhanced dramatically upon coordination in the presence of NO$_3$$^-$ anions, which assist deprotonation. Ring formation is catalyzed by Cu$^{II}$ in the second deprotonation step, i.e., Cu$^{II}$ accepts the negative charge of the deprotonated ligand, to assist in the formation of the neutral aromatic furoxan ring; in the

last step, Cu$^I$ is oxidized to Cu$^{II}$ by soluble oxygen, $O_2(MeOH)_n$, to retrieve its oxidation state and give the complex. In accordance with the proposed mechanistic scheme, **3** could not be prepared in the absence of oxygen (inert atmosphere). This indicates that the presence of molecular oxygen from the atmospheric air as an oxidant is essential for the formation of the furoxan ring.

**Scheme 2.** Synthesis of complex [Cu(NO$_3$)$_2$(L)] (**3**) [53]. Coordination bonds are indicated in bold.

## 2.2. Description of Structure

The structure of **1** was determined by single-crystal X-ray crystallography. Crystallographic data are listed in Table 1. Various structural plots are shown in Figures 3–6. Selected structural distances and angles are given in Table 2, while hydrogen bonding details are summarized in Table 3.

**Table 1.** Crystallographic data and structural refinement parameters for complex **1**.

| Parameter | [ZnCl$_2$(L)] (1) |
|---|---|
| Empirical formula | C$_{12}$H$_8$ZnCl$_2$N$_4$O$_2$ |
| Formula weight | 376.49 |
| Crystal system | monoclinic |
| Space group | $P2_1/n$ |
| Color | pale yellow |
| Crystal size, mm | 0.34 × 0.10 × 0.09 |
| Crystal habit | parallelepiped |
| $a$, Å | 6.4893(1) |
| $b$, Å | 14.9997(3) |
| $c$, Å | 14.2753(3) |
| $\alpha$, ° | 90.00(1) |
| $\beta$, ° | 90.637(1) |
| $\gamma$, ° | 90.00(1) |
| Volume, Å$^3$ | 1389.44(5) |
| Z | 4 |
| Temperature, K | 160 |
| Radiation, Å | Cu K$\alpha$, 1.54178 |
| Calculated density, g·cm$^{-3}$ | 1.800 |
| Absorption coefficient, mm$^{-1}$ | 6.09 |
| No. of measured, independent and observed [$I > 2\sigma(I)$] reflections | 15,854, 2338, 1890 |
| $R_{int}$ | 0.071 |
| Number of parameters | 191 |
| Final $R$ indices [$I > 2\sigma(I)$] [a] | $R_1 = 0.0705$, $wR_2 = 0.1514$ |
| Goodness-of-fit in $F^2$ | 1.12 |
| Largest differences peak and hole (e·Å$^{-3}$) | 0.73/−0.60 |

[a] $R_1 = \Sigma(|F_o|-|F_c|)/\Sigma(|F_o|)$, $wR_2 = \{\Sigma[w(F_o^2-F_c^2)^2]/\Sigma[w(F_o^2)^2]\}^{1/2}$, $w = 1/[\sigma^2(F_o^2) + (\alpha P)2 + bP]$, where $P = [\max(F_o^2, 0) + 2F_c^2]/3$ (a = 0.0408 and b = 8.4458).

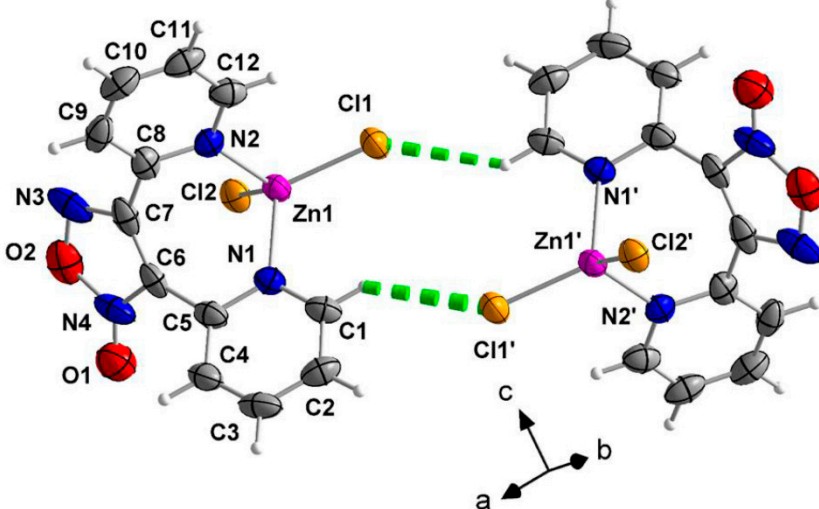

**Figure 3.** Two centrosymmetrically related hydrogen-bonded [ZnCl$_2$(L)] molecules that are present in the crystal structure of **1**. The thermal ellipsoids are presented at the 50% level. The light green dashed lines indicate the C1–H1···Cl1′ and C1′–H1′···Cl1 hydrogen bonds. The hydrogen atoms are not labelled. Symmetry codes: (′) −$x$, −$y$ + 2, −$z$ + 1.

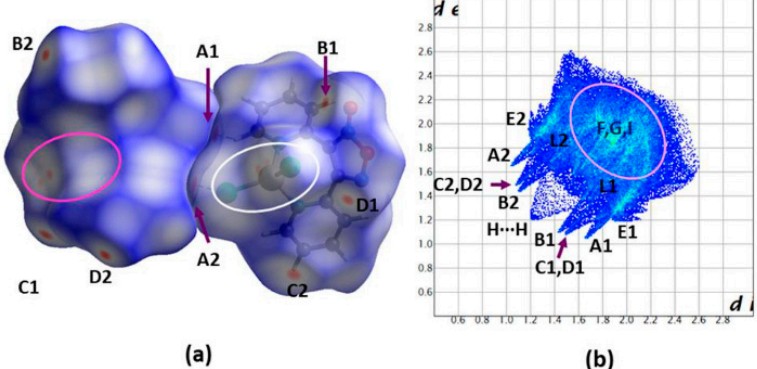

**Figure 4.** Hirshfeld surface (HS) analysis plots for complex [ZnCl$_2$(L)] (**1**): (**a**) HS decorated with the $d_{norm}$ property. A1, B1, C1 and D1 points indicate the areas of the acceptor atoms and A2, B2, C2 and D2 the areas of the hydrogen atoms of the donors in the C1–H1···Cl1′, C10–H10-O‴, C11–H11–O1$^{ii}$ and C3–H3–N3$^{i}$ hydrogen bonds, respectively. (**b**) Fingerprint (FP) plot for all types of interactions, with labels indicating the contribution of each type of interactions (see text for details).

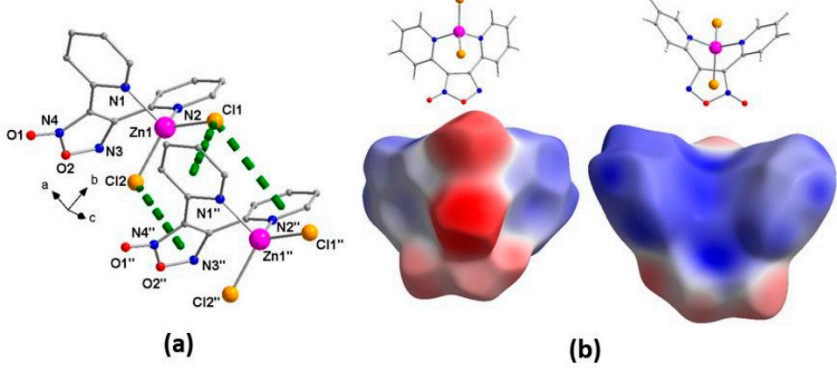

**Figure 5.** (**a**) Cl···$\pi$ intermolecular interactions, indicated with thick, green dashed lines, in the crystal structure of [ZnCl$_2$(L)] (**1**). Symmetry code: (″) $x$ − 1, $y$, $z$. Hydrogen atoms have been omitted for clarity. (**b**) Front and back views of the HS decorated electrostatic potentials in the range from −0.093 (red) to +0.066 (blue).

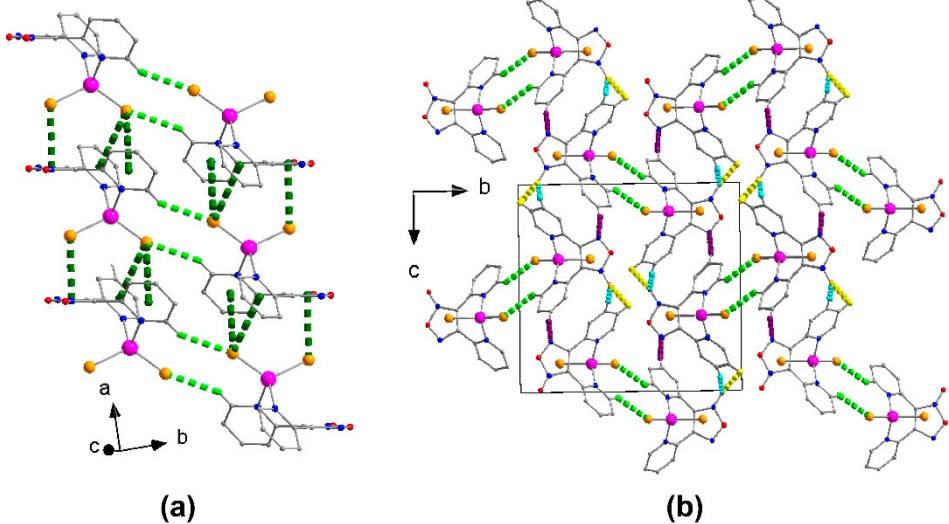

**Figure 6.** (**a**) A stack of hydrogen-bonded dimers of complex [ZnCl$_2$(L)] (**1**) along the *a* axis, formed through Cl···π intermolecular interactions indicated with thick, green dashed lines. (**b**) View along the *a* axis illustrating the stacks of hydrogen-bonded dimers and their interactions through C10–H10···O1, C11–H11···O1 and C3–H3···N3 interactions indicated with thick cyan, yellow and violet dashed lines, respectively. The intradimer C1–H1···Cl1 interactions are presented with light green dashed lines, while the Cl···π interactions and the hydrogen atoms not involved in hydrogen bonds are not shown for clarity.

**Table 2.** Selected bond lengths (Å) and angles (°) for complex [ZnCl$_2$(L)] (**1**).

| Bond Lengths (Å) | | Bond Angles (°) | |
|---|---|---|---|
| Zn1–Cl1 | 2.195(2) | Cl1–Zn1–Cl2 | 120.4(1) |
| Zn1–Cl2 | 2.221(2) | Cl1–Zn1–N1 | 111.7(2) |
| Zn1–N1 | 2.054(6) | Cl1–Zn1–N2 | 108.5(2) |
| Zn1–N2 | 2.068(6) | Cl2–Zn1–N1 | 107.3(2) |
| C6–C7 | 1.409(11) | Cl2–Zn1–N2 | 110.5(2) |
| C6–N4 | 1.341(11) | N1–Zn1–N2 | 95.8(2) |
| N4–O2 | 1.427(10) | N3–O2–N4 | 112.7(7) |
| N3–O2 | 1.334(9) | O2–N4–C6 | 107.1(6) |
| N3–C7 | 1.397(11) | O1–N4–O2 | 120.4(8) |
| N4–O1 | 1.170(8) | O1–N4–C6 | 132.5(9) |

**Table 3.** Hydrogen bonding interactions (Å, °) in the crystal structure of [ZnCl$_2$(L)] (**1**).

| D–H···A | d(D···A) | d(H···A) | <DHA | Symmetry Code of A |
|---|---|---|---|---|
| C1–H1···Cl1′ | 3.529(5) | 2.757(2) | 138.5(4) | −*x*, −*y* + 2, −*z* + 1 |
| C10–H10···O1‴ | 3.50(1) | 2.621(7) | 154.0(5) | *x* + 0.5, −*y* + 1.5, *z* + 0.5 |
| C3–H3···N3ⁱ | 3.57(1) | 2.658(7) | 162.0(5) | *x* + 0.5, −*y* + 1.5, *z* − 0.5 |
| C11–H11···O1ⁱⁱ ᵃ | 3.11(1) | 2.902(7) | 93.9(5) | *x*−0.5, −*y* + 1.5, *z* + 0.5 |

ᵃ Extremely weak (if any) hydrogen bond. Donor (D); acceptor (A).

The crystal structure of **1** consists of [ZnCl$_2$(L)] complex molecules. The Zn$^{II}$ atom is coordinated by two chloro (or chlorido) groups (Cl1, Cl2) and two 2-pyridyl nitrogen atoms (N1, N2), the latter two arising from the $N_{pyridyl},N'_{pyridyl}$-bidentate chelating ligand L. The two nitrogen donor atoms are located *cis* to each other, allowing the formation of the unusual seven-membered chelating ring. The coordination geometry of the metal ion is distorted tetrahedral, the donor atom–Zn$^{II}$–donor atom bond angles being in the range of 95.8(2)–120.4(1)°. The distortion from the regular tetrahedral geometry is primarily attributed to the small bite angle [95.8(2)°] of the seven-membered chelating ring.

The chelating ring is highly puckered, with the planes of the coordinated pyridyl rings being inclined to the furoxan ring at angles of 52.4(2) and 51.7(2)°, and mutually inclined at an angle of 73.7(2)°.

The Zn–Cl bond distances of 2.195(2) and 2.221(2) Å are in the expected range for metal complexes with $\{Zn^{II}Cl_2N_2\}$ distorted tetrahedral environments [56]. The Zn–N bond lengths [2.054(6) and 2.068(6) Å] are comparable to those observed for the tetrahedral complex [Zn(MNF)$_2$(L″)], where L″ is the 1,10-phenanthrolinefuroxan ligand and MNF$^-$ is the mefenamate(-1) group [55]. The furoxan bond lengths N3–O2, N4–O2 and N4–O1 are 1.334(9), 1.427(10) and 1.170(8) Å, respectively, in agreement with those measured for other metal complexes containing a furoxan ring [47,49,51–53]. Noteworthy structural features are the long N4–O2 and short N4–O1 (exocyclic) bonds, the latter indicating an appreciable double-bond character in the resonance hybrid of the coordinated L [49].

Complex **1** contains several different atoms, resulting in a plethora of intermolecular interactions and thus the Hirshfeld Surface (HS) analysis tool [57] is valuable in revealing the role of each type of interaction in the building of the crystal structure. The molecules form centrosymmetrically related dimers through C1–H1···Cl1′ (and symmetry equivalent) hydrogen bonds (Figure 3) with parameters in the expected range [58–60]. Systematic studies [61] have shown that, under the influence of primarily weak C–H···Cl hydrogen bonding, neutral $M^{II}Cl_2$-containing (M = divalent first-row transition metal) coordination modules, supported by chelating ligands, act as supramolecular synthons of variable metal coordination geometries to construct supermolecules with interesting networks. The HSs for the two molecules forming the dimer, decorated with the $d_{norm}$ property in the orientation shown in Figure 3, with one of them in transparency mode, are presented in Figure 4a. Figure 4b presents the fingerprint (FP) plot with the contribution from all types of interactions; these related hydrogen bonds, Cl···H/H···Cl, O···H/H···O and N···H/H···N, contribute with 21.6, 16.7 and 12.5%, respectively. The pairs of red areas labelled (A1,A2), (B1,B2), (C1,C2) and (D1,D2) correspond to the hydrogen bonds listed in Table 3.

The HS analysis also reveals the intermolecular interactions of neighboring molecules through Cl···π interactions [62–64], see Figure 5, which result in the formation of stacks along the $a$ axis, Figure 6a. The distances Cl1···Cg1″, Cl1···Cg2″ and Cl2···Cg3″ are 3.832(2), 3.874(2) and 3.183(2) Å, respectively; Cg1″, Cg2″ and Cg3″ are the centroids of the rings defined by the atoms (N1″, C1″, C2″, C3″, C4″, C5″), (N2″, C8″, C9″, C10″, C11″, C12″) and (C6″, C7″, N3″, O2″, N4″), respectively (symmetry code: (″) $x - 1, y, z$). The areas on the $d_{norm}$ decorated surface, where these types of interactions contribute, are indicated with a white ellipse (acceptor area) and a pink one (donor area), see Figure 4a. These parts of HS correspond to Cl···C/C···Cl, Cl···N/N···Cl and Cl···O/O···Cl types of interactions (labelled as F, G and I in Figure 4b) and they contribute 8.7, 2.8 and 2.0%, respectively, in the FP plot of interactions in **1** in the area indicated with an ellipse in Figure 4b. All the above-mentioned types of interactions (hydrogen bonds and Cl···π) add up to a total of 64.3% contribution in the FP plot. The remaining interactions (35.7%) are the H···H (12.2%), C···H (17.1%, labelled as E1, E2 in the characteristic wing-shaped areas for these types of interactions, Figure 4b), C···O/O···C (28% at the points indicated with labels L1 and L2 in Figure 4b), plus other types of interactions with distances lower than the sum of their van der Waals radii. The complementarity of the parts of the contacts of neighboring molecules indicated with ellipses in Figure 4a are also supported by the HS mapped with electrostatic potential values calculated through Crystal Explorer [57,65], see Figure 5b. The hydrogen-bonded dimers stack along the $a$ axis, as shown in Figure 6a; the stacks further interact through the hydrogen bonds discussed above and listed in Table 3, building the 3D architecture of the complex (Figure 6b). For the C11–H11–O1$^{ii}$ bond, the H11···O1$^{ii}$ distance is long [2.902(7) Å] and the corresponding C11–H11–O1$^{ii}$ angle is small [93.5(5)°], see Table 3. Since the C11···O1$^{ii}$ distance is rather short [3.11(1) Å], it seems that a C···O/O···C type of interaction [66] is present.

## 2.3. Spectroscopic Characterization in Brief

The IR spectrum of complex **1** (Figure S1) is complicated and makes assignments of the bands difficult. The strongest band is located at 1604 cm$^{-1}$ and can be assigned to two vibrational modes, namely the

stretching vibration of the double carbon-nitrogen bond, $v$(C=N), of the C=N–O grouping [49,67] and a 2-pyridyl stretch [53]. These vibrations appear as two distinct peaks at 1604 and 1597 cm$^{-1}$ in the Raman spectrum of the compound (Figure S2) [68]. The bands at 1418 and 684 cm$^{-1}$ are assigned to the $v$(C=N) vibration of the C=NO$_2$ grouping [49] and to an in-plane deformation of the 2-pyridyl groups [69], respectively. The Raman peaks at 3066 and 1026 (which is the strongest in the spectrum) are due to an aromatic C–H stretch and the "breathing" mode of the 2-pyridyl rings [68], respectively.

The $^1$H NMR spectrum of **1** in $d_6$-DMSO (Figure S3) indicates decomposition, i.e., the non-retainment of the structure, in solution. The most remarkable feature is that the spectrum is identical to the spectrum of the free ligand, as the latter was obtained under identical conditions and as reported in the literature [52]. No coordination-induced shifts were observed. Richardson and Steel have fully assigned the signals in the $^1$H NMR spectrum of L in $d_6$-DMSO by means of 1D-TOCSY experiments [52]. Based on their study and following (for clarity) the numbering scheme of L in the molecule of the complex (Figure 3, left), the signals at $\delta$ values of 8.56, 8.01, 7.91, 7.87, 7.56 and 7.50 ppm in Figure S3 are assigned to protons H1/H12, H3/H10, H9, H4, H11 and H2, respectively. The general trend that the exocyclic oxygen of furoxans exerts a shielding influence on protons of substituents attached to C6 [49] is clearly observed here.

The above-mentioned experimental fact indicates that the complex decomposes, most probably as indicated by equation (1), where $x = 4$–6. Strong evidence of our proposal comes from the molar conductivity value, $\Lambda_M$ ($10^{-3}$ M, 25 °C), for the complex in DMSO, which is 75 S cm$^2$ mol$^{-1}$, indicative of a 1:2 electrolyte [70]. This conclusion is further supported by the $^{13}$C and $^{15}$N NMR spectra of **1** in the same solvent. Each spectrum is identical to the corresponding spectrum of L. The $^{13}$C NMR spectrum of **1** (Figure S4) shows the twelve expected signals. Adopting again the numbering scheme of L in Figure 3, the C6 and C7 resonances of the central 1,2,5-oxadiazole-2-oxide ring are at $\delta$ 114.9 and 156.9 ppm, respectively [49], while the ten signals of the 2-pyridyl groups are in the $\delta$ 124.3–150.2 ppm region [71,72]. The two 1,2,5-oxadiazole ring nitrogen atoms in **1** (the $^{15}$N NMR spectrum in $d_6$-DMSO is of poor quality) appear as distinct signals at $\delta$ values of approximately $-20$ and $-3$ ppm for N4 and N5, respectively [49], while the two 2-pyridyl nitrogen atoms display signals at $\delta$ $-75.5$ and $-52.5$ ppm [73]. Exact assignments for the 2-pyridyl $^{13}$C and $^{15}$N resonances would be risky [74].

$$[\text{ZnCl}_2(\text{L})] + x \, d_6 - \text{DMSO} \xrightarrow{d_6 - \text{DMSO}} [\text{Zn}(d_6 - \text{DMSO})_x]^{2+} + 2\text{Cl}^- + \text{L} \tag{1}$$

Upon maximum excitation at 375 nm, solid complex **1** shows an emission band with a maximum at 452 nm (emission of blue light) Figure 7. As Zn(II) is non-emissive, the emission can be assigned to a charge-transfer state within the coordinated ligand [75,76]. Additional evidence for the intraligand character of the blue emission in the complex comes from the study of the emission properties of the free ligand L in the solid state and in solution (Figure S5); the solid-state and solution emission spectra of L are almost identical. Upon maximum excitation at 373 nm, the ligand L emits at 479 nm. The difference in the emission $\lambda_{max}$ between **1** and L may be ascribed to coordination.

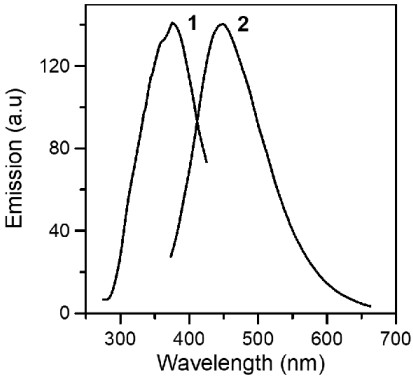

**Figure 7.** Solid-state, room-temperature excitation (curve 1; maximum emission at 452 nm) and emission (curve 2; maximum excitation at 375 nm) spectra of complex [ZnCl$_2$(L)] (**1**).

### 2.4. Theoretical Studies and Mechanistic Aspects

The geometric and energetic profiles of possible pathways involved in the reaction of ClpaoH and $Zn(NO_3)_2·6H_2O$ in solution (MeOH) in the presence of $Et_3N$ were explored by DFT computational methodologies. The enthalpy changes ($\Delta H$ in Kcal/mol) of all possible reactions calculated by the PBE0/Def2-TZVP(Cr)∪6-31G(d,p)(E)/Polarizable Continuum Model (PCM) computational protocol are given in Table 4.

**Table 4.** Thermodynamics of possible reaction pathways involved in the reaction of ClpaoH with $Zn(NO_3)_2·6H_2O$ in solution (MeOH) in the presence of $Et_3N$ calculated at the PBE0/Def2-TZVP(Cr)∪6-31G(d,p)(E)/Polarizable Continuum Model (PCM) level of theory.

| Reaction | $\Delta H$ (Kcal/mol) |
|---|---|
| ClpaoH → paoH$^+$ + Cl$^-$ | 43.8 |
| ClpaoH + Et$_3$N → Clpao$^-$ + Et$_3$NH$^+$ | 5.5 |
| Clpao$^-$ → pao + Cl$^-$ | 8.2 |
| 2 Clpao$^-$ + Zn$^{2+}$ → 2pao + ZnCl$_2$ | −84.7 |
| Clpao$^-$ + Zn$^{2+}$ → [Zn(Clpao)]$^+$ | −34.3 |
| Clpao$^-$ + Zn$^{2+}$ → [ZnCl(pao)]$^+$ | −63.0 |
| 2 pao + ZnCl$_2$ → complex 1 | −48.2 |

The estimated bond dissociation energy (BDE) of the C–Cl bond in ClpaoH is 43.8 Kcal/mol which decreases remarkably to 8.2 Kcal/mol after the deprotonation of the ligand. The deprotonation of ClpaoH by reaction with $Et_3N$ demands only 5.5 Kcal/mol. The rupture of the C–Cl bond of the deprotonated Clpao$^-$ ligand by interaction with the metal ion to afford the pao molecule (this is a nitrile oxide) and $ZnCl_2$ is a strongly exothermic process ($\Delta H = −84.7$ Kcal/mol). Scheme 3 illustrates the natural atomic charge distribution and the Frontier Molecular Orbitals of the pao ligand, which undergoes C–C coupling upon coordination to $ZnCl_2$ (through the 2-pyridyl nitrogen atom), yielding the intermediate $[ZnCl_2(pao)_2]$ complex. The formation of this intermediate is slightly exothermic ($\Delta H = −13.6$ Kcal/mol). A cross-coupling reaction takes place, affording the final Zn(II) complex **1**. The geometric and energetic reaction profiles for the cross-coupling reaction between the coordinated pao ligands promoted by the $[ZnCl_2(pao)_2]$ complex, calculated at the PBE0/Def2-TZVP(Cr)∪6-31G(d,p)(E)/PCM level of theory in MeOH solution, are depicted in Figure 8.

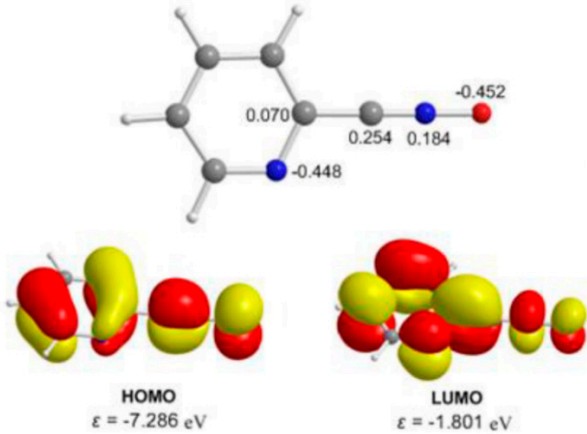

**Scheme 3.** Natural atomic charges and Frontier Molecular Orbitals of the nitride oxide pao ligand.

The course of the reaction involves, initially, the deprotonation and dechlorination of ClpaoH, yielding the pao ligands which are weakly associated to $ZnCl_2$ and forming the tetrahedral $[ZnCl_2(pao)_2]$ intermediate, followed by a cross-coupling reaction, yielding the final product **1** (pathway A in Figure 8). The cross-coupling reaction proceeds through a transition state (TS), surmounting a low

activation barrier of 23.0 Kcal/mol. The normal coordination vectors (arrows) of the vibrational modes, corresponding to the imaginary frequency ($v_i$ = 447.9 cm$^{-1}$) of the TS, are consistent with the cross-coupling intermediacy. The complete course of the reaction is predicted to be exothermic, with the estimated enthalpy changes ΔH being −132.7 Kcal/mol. The total ΔG change is −130.8 Kcal/mol, indicative of the spontaneous exergonic reaction.

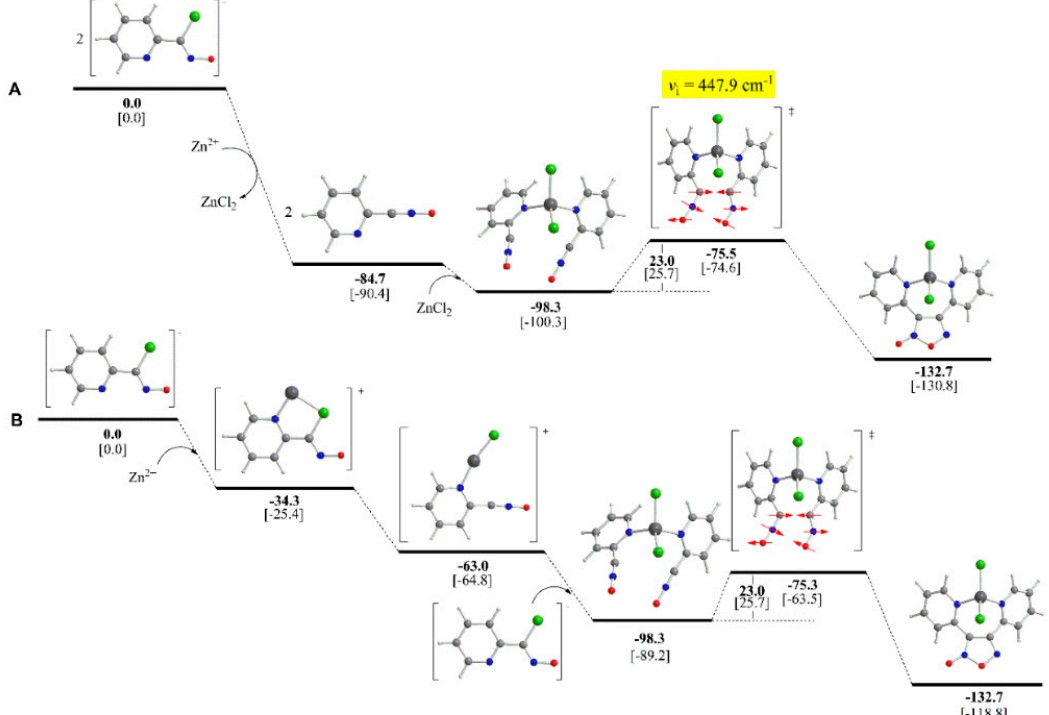

**Figure 8.** Geometric and energetic (ΔH in Kcal/mol, bold numbers/ΔG in Kcal/mol, plain numbers in brackets) reaction profile for the cross-coupling reaction between the coordinated pao ligands promoted by the [ZnCl$_2$(pao)$_2$] complex at the PBE0/Def2-TZVP(Cr)∪6-31G(d,p)(E)/PCM level of theory in solution (MeOH).

Alternatively, the deprotonated Clpao$^-$ ligand could undergo dechlorination through coordination to Zn$^{2+}$ or solvated Zn$^{2+}$ ions, yielding the [Zn(Clpao)]$^+$ intermediate. This promotes the rupture of the C–Cl bond through the coordination of the Cl atom to Zn$^{2+}$, yielding the more stable (by 28.7 Kcal/mol) [ZnCl(pao)]$^+$ intermediate. The formation of [Zn(Clpao)]$^+$ is exothermic (ΔH = −34.3 Kcal/mol). Next, the intermediate [ZnCl(pao)]$^+$ reacts with a second Clpao$^-$ ion, yielding the [ZnCl$_2$(pao)$_2$] intermediate, which promotes the cross-coupling reaction (pathway B in Figure 8). The total ΔG value for path B is −118.8 Kcal/mol, also indicative of the spontaneous exergonic reaction.

It is important to note that the cross-coupling of the pao species is a new type of cross-coupling reaction, which proceeds in the absence of any transition metal catalyst, as in the case of the Negishi [77], Heck [78], Stille [79], Suzuki [80], Sonogashira [81] and Buchwald-Hartwig [82,83] cross-coupling reactions. It is promoted by the coordination of the pao species to the {ZnCl$_2$} entity, resulting from the dechlorination of Clpao$^-$ by Zn$^{2+}$ or solvated Zn$^{2+}$ ions.

The structures of the ligands, intermediates, TS and product optimized at the PBE0/Def2-TZVP(Cr)∪6-31G(d,p)(E)/PCM level of theory in MeOH solution are shown in Figure 9. This figure clearly shows the elongation of the C–Cl bond of ClpaoH by 0.039 Å after its deprotonation by the Et$_3$N base. The C–Cl bond is further elongated by 0.040 Å upon coordination of Clpao$^-$ to Zn$^{2+}$ or the solvated Zn$^{2+}$ ion, leading to the rupture of the bond and the formation of the [ZnCl(pao)]$^+$ intermediate, which contains a chloro ligand. In the tetrahedral [ZnCl$_2$(pao)$_2$] intermediate and the TS, the nitrile oxide pao ligands are coordinated to the {ZnCl$_2$} moiety through the 2-pyridyl nitrogen

atoms. Noteworthily, the optimized equilibrium geometry of the final product **1** is compatible with the solid-state geometry determined by single-crystal X-ray crystallography (Table 2). However, we note the expected slight elongation in almost all bond lengths in solution relative to the solid-state structure.

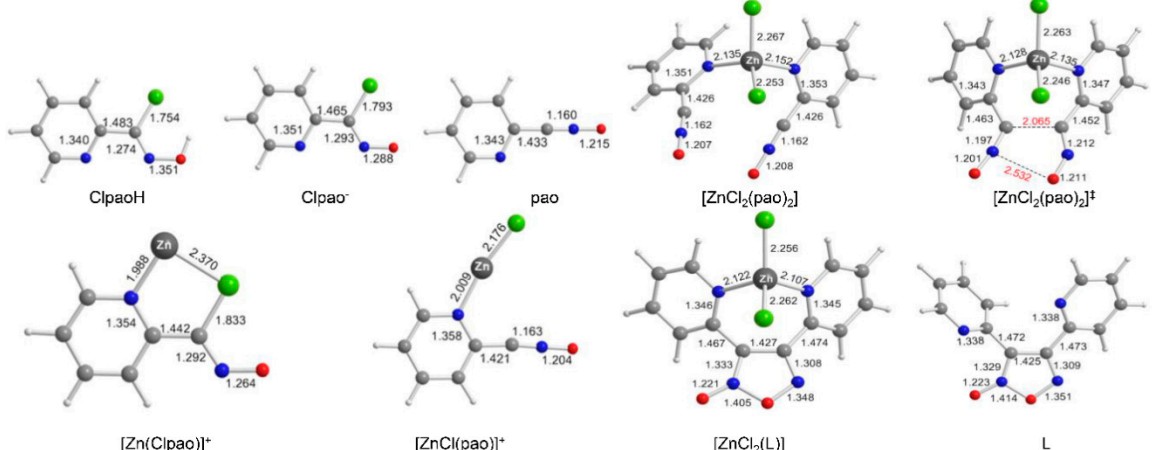

**Figure 9.** Equilibrium geometries of the ligands, intermediates, transition state (TS) and product in MeOH solution optimized at the PBE0/Def2-TZVP(Cr)∪6-31G(d,p)(E)/PCM level of theory.

## 3. Materials and Methods

### 3.1. Materials

All manipulations were performed under aerobic conditions. Reagents and solvents were purchased form Alfa Aesar (Karlsruhe, Germany) and Sigma-Aldrich (Tanfrichen, Germany) and used as received. The compound pyridine-2-amidoxime, $(NH_2)$paoH, was synthesized as described in the literature [84]. Pyridine-2-chloroxime (or *N*-hydroxy-2-pyridinecarboximidoyl chloride), ClpaoH, was synthesized as follows (Scheme 1): an amount of 13.69 g (99.8 mmol) of $NH_2$paoH was dissolved in a mixture of concentrated hydrochloric acid (100 mL) and $H_2O$ (200 mL). The solution was cooled to −5 °C and then a solution of $NaNO_2$ (8.00 g, 115.9 mmol) in $H_2O$ (40 mL) was added dropwise. The reaction mixture was stirred at −5 °C for 2 h. The solution obtained was filtered and to the filtrate a saturated aqueous solution of $NaHCO_3$ was added until a pH = 3 was reached. A white to cream solid was precipitated, washed with ice-cold $H_2O$ (5 × 3 mL) and dried in a vacuum dessicator over $P_4O_{10}$. The pure compound was obtained after recrystallization from $Et_2O$. Yield: 75%. Anal. Calcd. (%) for $C_6H_5N_2OCl$: C, 46.02; H, 3.23; N, 17.90. Found (%): C, 45.88; H, 3.33; N, 18.13. IR (KBr, cm$^{-1}$): 3098m, 2994m, 2786mb, 1734wb, 1588m, 1480s, 1430sh, 1384sh, 1302m, 1264w, 1232m, 1154w, 1110w, 1086sh, 1016s, 954s, 898sh, 870mb, 774s, 736m, 688m, 651sh, 618m, 572m, 552w, 468w, 426sh, 402m. $^1$H NMR ($d_6$-DMSO, $\delta$/ppm, see Figure S6): 12.65 (s, 1H), 8.71 (d, 1H), 7.88 (mt, 2H), 7.48 (t, 1H). $^{13}$C NMR ($d_6$-DMSO, $\delta$/ppm, See Figure S7): 150.2, 149.8m, 137.6, 137.2, 125.4, 122.0. Upon maximum excitation at 317 nm, the solid compound and its methanolic solutions show an emission band with a maximum at 369 nm (Figure S8).

The free ligand di(2-pyridyl)furoxan, L, was synthesized in three steps from 2-pyridylaldoxime, paoH, as described in the literature [43,52]. Yield: 35%. Its purity was confirmed by microanalyses, melting point determination, and $^1$H and $^{13}$C NMR spectroscopy. Anal. Calcd. (%) for $C_{12}H_8N_4O_2$: C, 59.99; H, 3.36; N, 23.33. Found (%): C, 60.37; H, 3.28; N, 23.01. Melting point (after recrystallization from water): 167–169 °C (literature value [43]: 169–170.5 °C). $^1$H NMR ($d_6$-DMSO, $\delta$/ppm): 8.56 (t, 2H), 8.01 (mt, 2H), 7.91 (d, 1H), 7.86 (d, 1H), 7.56 (q, 1H), 7.51 (q, 1H). $^{13}$C NMR ($d_6$-DMSO, $\delta$/ppm): 156.9, 150.2, 150.1, 146.4, 143.6, 137.9, 137.7, 126.1, 125.6, 125.2, 124.3, 114.9. Upon maximum excitation at 373 nm, an emission with a maximum at 479 nm (in both solid state and MeOH solution) was recorded.

### 3.2. Physical and Spectroscopic Measurements

Elemental analyses (C, H, N) were performed by the University of Patras Center of Instrumental Analysis. Zinc was determined by decomposing the samples in dilute aqueous ammonia and titrating with standard 0.05 M EDTA solution using Eriochrome Black T as the indicator and a buffer solution of pH = 10 ($NH_3$-$NH_4Cl$). For the chloride determination, the samples were decomposed in 2N $HNO_3$. The quantity of $Cl^-$ was analyzed by potentiometric titration with a standard 0.1 N $AgNO_3$ solution using a Corning Eel model 12 potentiometer, and calomel and sulfide electrodes. Conductivity measurements were carried out at 25 °C with a Metrohm-Herisau E-527 bridge (Herisau, Switzerland) and a cell of standard constant. FT-IR spectra were recorded using a PerkinElmer 16PC spectrometer (PerkinElmer, Waltham, MA, USA) with samples prepared as KBr pellets. FT Raman spectra were obtained using a Bruker (D) FRA-106/S component (Bruker, Karlsruhe, Germany) attached to an EQUINOX 55 spectrometer. An R510 diode-pumped Nd: YAG polarized laser at 1064 nm (with a maximum output power of 500 mW) was used for Raman excitation in a 180° scattering sample illumination module. Optical filtering reduced the Rayleigh elastic scattering and, in combination with a $CaF_2$ beam splitter and a high-sensitivity liquid $N_2$-cooled Ge detector, allowed the Raman intensities to be recorded in the Stokes-shifted Raman region, all in one spectrum. The spectrum shown in Figure 7 is an average of 1000 scans at 4 $cm^{-1}$ resolution; the intensity of the Nd:YAG on the sample was 100 mW. Solid-state and solution emission and excitation spectra were recorded using a Cary Eclipse fluorescence spectrometer (Varian, Palo Alto, CA, USA) at room temperature. $^1H$, $^{13}C$ and $^{15}N$ NMR spectra in $d_6$-DMSO were recorded on a 600.13-MHz Bruker Avance DPX spectrometer (Bruker Avance, Billerica, MA, USA).

### 3.3. Syntheses of the Complex

Method (a): To a stirred, colorless solution of $Zn(NO_3)_2 \cdot 6H_2O$ (0.030 g, 0.10 mmol) in MeOH (10 mL) we added $Et_3N$ (28 μL, 0.20 mmol) and solid ClpaoH (0.031 g, 0.20 mmol). The solid soon dissolved and the resulting pale yellow solution was stirred for a further 10 min, filtered and stored in a closed flask at room temperature. X-ray-quality pale yellow crystals of the product were obtained within a period of 2 d. The crystals were collected by filtration, washed with ice-cold MeOH (1 mL) and $Et_2O$ (2 × 3 mL), and dried in air. Yield: 61%. Anal. Calcd. (%) for $C_{12}H_8N_4ZnCl_2O_2$: C, 38.28; H, 2.15; N, 14.88; Zn, 17.37; Cl, 18.83. Found (%): C, 38.11; H, 2.20; N, 14.57; Zn, 17.52; Cl, 18.71. IR (KBr, $cm^{-1}$): 1604s, 1568m, 1528w, 1504m, 1468w, 1418m, 1342w, 1292w, 1164w, 1146m, 1094m, 1060m, 1028m, 1008w, 974w, 832w, 796sh, 780s, 756w, 684m, 644w, 548w, 520w, 422w. Raman ($cm^{-1}$):3089w, 3066m, 1604s, 1597m, 1570m, 1539m, 1505m, 1462m, 1443w, 1418w, 1372w, 1343m, 1321w, 1254w, 1196m, 1163m, 1147w, 1103w, 1095w, 1059w, 1026s, 1007w, 982w, 974w, 903w, 834w, 795w, 780w, 753w, 737w, 720w, 687w, 648w, 602w, 548w, 519w, 426w. $^1H$ NMR ($d_6$-DMSO, δ/ppm): 8.56 (t, 2H), 8.01 (mt, 2H), 7.91 (d, 1H), 7.85 (d, 1H), 7.56 (q, 1H). $^{13}C$ NMR ($d_6$-DMSO, δ/ppm):156.9, 150.2, 150.1, 146.4, 143.6, 137.9, 137.7, 126.1, 125.6, 125.2, 124.3, 114.9. $^{15}N$ NMR ($d_6$-DMSO, δ/ppm-approximate values): −75.5, −52.5, −20.0, −3.0. $\Lambda_M$ (DMSO, $10^{-3}$ M, 25 °C): 75 S $cm^2$ $mol^{-1}$. Upon maximum excitation at 375 nm, a solid sample of **1** emits blue light with a maximum at 452 nm.

Method (b): To a stirred, colorless solution of $ZnCl_2$ (0.014g, 0.10 mmol) in MeOH (5 mL) was added solid L (0.024 g, 0.10 mmol). The solid soon dissolved and the resulting pale yellow solution was stirred overnight at room temperature. The resulting pale yellow microcrystalline powder was collected by filtration, washed with ice-cold MeOH (2 × 0.5 mL) and $Et_2O$ (4 × 1 mL), and dried in a vacuum dessicator over anhydrous $CaCl_2$. Yield: ~70%. The IR, Raman and $^1H$ NMR ($d_6$-DMSO) spectra of the sample are identical to the corresponding spectra of authentic **1** prepared by method (a).

### 3.4. Single-Crystal X-Ray Crystallography

A pale yellow crystal of **1** (0.09 × 0.10 × 0.34 nm) was taken from the mother liquor and immediately cooled to 160 K (−113 °C). Diffraction data were collected on a Rigaku R-Axis Image Plate diffractometer

(Rigaku, Americas Corporation, The Woodlands, TX, USA) using graphite-monochromated Cu Kα radiation. Data collection (ω-scans) and processing (cell refinement, data reduction and empirical absorption correction) were performed using the CrystalClear program package [85]. The structures were solved by direct methods using SHELXS, ver. 2013/1 [86] and refined by full-matrix least squares techniques on $F^2$ with SHELXL, ver. 2014/6 [87]. As angle β (90.637(1)°) is close to 90°, the crystal presents pseudomerohedral twinning with a twin law (−1, 0, 0/0, 1, 0/0, 0, 1) and the BASF parameter was converged to the 0.031(2) value. All non-H atoms were refined anisotropically. All H atoms were introduced at calculated positions and refined as riding on their corresponding bonded atoms. Plots of the structures were drawn using the Diamond 3 program package [88]. Further crystallographic details for **1**: $2\theta_{max} = 130°$, $(\Delta/\sigma)_{max} = 0.002$, $R_1/wR_2$ (for all data) = 0.0858/0.1654.

Crystallographic data were deposited with the Cambridge Crystallographic Data Center, No. 2021375. Copies of the data can be obtained free of charge upon application to CCDC, 12 Union Road, Cambridge, CB2 1EZ, UK: Telephone: +(44)-1223-336033; E-mail: deposit@ccdc.ac.uk, or via https://www.ccdc.cam.ac.uk/structures/.

### 3.5. Computational Details

All calculations were performed using the Gaussian 09, version D.01, program suite [89]. The geometries of the complexes were fully optimized without symmetry constraints, employing the 1999 hybrid functional of Perdew, Burke and Ernzerhof [90–92], as implemented in the Gaussian 09, version D.01, program suite. This functional uses a 25% exchange and a 75% correlation weighting and is denoted as PBE0. The PBE0 functional was chosen because it is well established to be suitable for performing calculations of the geometric and energetic properties of transition metal complexes, providing good quality geometries and energies and the best agreement with high-level CCSD(T) computations [93]. Geometry optimization was performed in solution (MeOH solvent) using the Def2-TZVP basis set [94] for Zn and the 6-31+G(d) basis set for all main group elements (E). Solvent effects were accounted for by means of the Polarizable Continuum Model (PCM) using the integral equation formalism variant (IEF-PCM), this being the default Self-Consistent Reaction Field (SCRF) method [95]. The computational protocol used in DFT calculations was abbreviated as PBE0/Def2-TZVP(Cr)∪6-31G(d,p)(E)/PCM. All stationary points were identified as minima (number of imaginary frequencies NImag = 0). NBO population analysis was performed using Weinhold's methodology [96,97].

## 4. Concluding Comments and Perspectives

In this work, we show that the reaction of pyridine-2-chloroxime and hydrated zinc(II) nitrate in methanol led to complex [ZnCl₂(L)] (**1**) containing the 3,4-di(2-pyridyl)-1,2,5-oxadiazole-2-oxide or di(2-pyridyl)furoxan ligand (L). The chloro ligands arise from ClpaoH. The tetrahedral complex exhibits interesting supramolecular features and was fully characterized by three techniques (IR, Raman and photoluminescence spectroscopies) in the solid state. In DMSO solutions, **1** decomposes, releasing the free ligand, as proven by NMR spectroscopies ($^1$H, $^{13}$C and $^{15}$N NMR) and a conductivity measurement; unfortunately, the complex is not soluble in non-coordinating solvents, e.g., CHCl₃, which would offer more detailed solution studies.

The most salient feature of this work is the ClpaoH → L transformation during the reaction. Complex **1** is only the third structurally characterized compound containing L as a ligand. The two previously reported compounds [Cu₂Cl₄(L)₂] (**2**) [52] and [Cu(NO₃)₂(L)] (**3**) [53] (the latter has just been published) were prepared by completely different methods. The dinuclear complex was prepared by using the pre-synthesized ligand L, whereas the mononuclear compound was synthesized upon metal-assisted oxidation of the dioxime ligand L′H₂ (Scheme 2). In all three complexes, L behaves as a $N_{pyridyl}$,$N'_{pyridyl}$-bidentate chelating ligand, forming an unusual seven-membered ring.

From detailed DFT studies in MeOH (the solvent in which **1** was prepared), it is proposed that the formation of the Zn(II) complex occurs via an unprecedented, uncatalyzed cross-coupling reaction between two coordinated 2-pyridyl nitrile oxide ligands (pao), as shown in Figure 8. The presence of

a base (Et$_3$N) is critical, because otherwise **1** cannot be isolated. Since the free ligand L is prepared from metal ion-free media [43,52], the in situ generation of the coordinated L in **1** cannot be strictly considered as Zn$^{II}$-promoted/mediated. However, it seems that the Zn$^{II}$ center brings two 2-pyridyl nitrile oxide molecules (produced by dechlorination and deprotonation of ClpaoH) close enough, facilitating the ring formation of the 1,2,5-oxadiazole-2-oxide moiety, i.e., the cross-coupling reaction. Moreover, our study provides strong evidence of the belief that the C-functionalization of oximes proceeds via the generation of intermediate nitrile oxides [7].

With the valuable knowledge and experience obtained in this study, our research directions are numerous. Among our future goals are: (1) the study of the reactions between ClpaoH and other metal ions (Mn$^{II}$, Fe$^{II}$, Fe$^{III}$, Co$^{II}$, Ni$^{II}$, Cd$^{II}$, ... ) in the presence of an external base; (2) efforts to prepare metal complexes of the non-transformed ClpaoH or Clpao$^-$ ligands in the absence of an external base; and (3) attempts to isolate metal complexes of 2-pyridyl nitrile oxide (pao) and investigate their reactivity. The realization of the two latter goals will shed experimental light on the mechanisms discussed in this work. Our progress, which is already advancing well, will be reported soon.

**Supplementary Materials:** The following Supplementary Materials are available as at: http://www.mdpi.com/2304-6740/8/9/47/s1, Figure S1: The IR spectrum of **1**; Figure S2: The Raman spectrum of **1**; Figure S3: The $^1$H NMR spectrum of **1** ($d_6$-DMSO) in the aromatic region; Figure S4: The $^{13}$C NMR spectrum of **1** in $d_6$-DMSO; Figure S5: Excitation and emission spectra of L in MeOH; Figure S6: The $^1$H NMR spectrum of ClpaoH in $d_6$-DMSO; Figure S7: The $^{13}$C NMR spectrum of ClpaoH in $d_6$-DMSO; Figure S8: Excitation and emission spectra of L in MeOH. Table S1: Cartesian coordinates and selected energetic data for the species participating in the reaction of ClpaoH and Zn(NO$_3$)$_2$·6H$_2$O in MeOH in the presence of Et$_3$N; The CIF and the checkCIF output files are included in the Supplementary Materials.

**Author Contributions:** S.T.T. contributed towards the synthesis, crystallization and IR, Raman and NMR characterization of complex **1**; he also performed an extensive literature search, prepared some figures and typed the manuscript. D.I.T. synthesized the organic compounds ClpaoH and L, and contributed towards their spectroscopic characterization. V.B. recorded the room temperature, solid state, solution excitation and emission spectra of ClpaoH, L and **1**, and interpreted the results. C.P.R. and V.P. collected single-crystal X-ray crystallographic data for **1**, solved the structure and performed its refinement; the latter also studied in detail the supramolecular features of the crystal structure by performing the HS analysis, and wrote the relevant part of the paper. A.T. carried out the theoretical study and wrote the relevant part of the paper. V.P., A.T. and S.P.P. coordinated the research and wrote the paper based on the detailed reports of the collaborators. All the authors exchanged opinions concerning the progress of the project and commented on the writing of the manuscript at all stages. All authors have read and agreed to the published version of the manuscript.

**Funding:** V.P. would like to thank the special Account of the NCSR "Demokritos" for financial support concerning the operation of the X-ray facilities at INN through the internal program entitled "Structural study and characterization of crystalline materials" (NCSR "Demokritos", ELKE #10813).

**Acknowledgments:** The authors would like to thank G.A.V. and Z.G.L. for their helpful discussions concerning the Raman spectrum of **1**.

**Conflicts of Interest:** The authors declare no conflict of interest.

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
