# Peer review of "Reactivity of Coordinated 2-Pyridyl Oximes: Synthesis, Structure, Spectroscopic Characterization and Theoretical Studies of Dichlorodi{(2-Pyridyl)Furoxan}Zinc(II) Obtained from the Reaction between Zinc(II) Nitrate and Pyridine-2-Chloroxime"

_inorganics, doi:10.3390/inorganics8090047_

Round 1
Reviewer 1 Report
The authors report the synthesis, structure, spectroscopic characterization of a zinc(II) complex, namely [ZnCl2(L)], L = [3,4-di(2-pyridyl)-1,2,5-oxadiazole-2.oxide] ligand. The complex has been investigated by FT IR and Raman spectroscopy, single crystal X-ray structure investigation, luminescence spectroscopy, Hirshfeld surface analysis with fingerprint plots. Advanced DFT computational methodologies were used to analyse energetic profiles of possible reaction pathways. The study shows an unexpected cross coupling reaction between two coordinated 2-pyridyl nitrile oxide ligands.
The work was performed both experimetally and theoretically in a very careful manner. The manuscript is well written with 116 references cited.
I recommend acceptance after taking into account following very minor comments/recommendations:
line 576: author please check version number: ver. 1013/1
line 583: please replace "obtained" by "deposited"
supplementary section:
I recommend to add PXRD pattern of title complex for bulk phase purity check.
Author Response
line 576: author please check version number: ver. 1013/1
The comment is correct. The version in ‘’2013/1’’ and we have corrected this typo in the revised ms.
line 583: please replace "obtained" by "deposited"
We have incorporated the correct term in the revised ms.
supplementary section:
I recommend to add PXRD pattern of title complex for bulk phase purity check.
The comment is scientifically correct. However, the PXRD instrument of our University has been out-of-order (due to a rather heavy damage) since last February. Due to the pandemic (and also because of the financial crisis in Greece), the instrument has not been repaired yet. The bulk phase purity has been confirmed by a variety of techniques (mainly C, H, and N microanalyses, 1H, 13C and 15N NMR spectroscopy, …). It is also important to note that we have employed X-ray quality single crystals of the product (before we transform them to powder) for its detailed characterization. In addition, during this week we performed zinc(II) and chloride analyses of the product sample (the relevant information has been added in parts 3.2 and 3.3 of the revised ms) and the experimental values are very close to the calculated (theoretical) ones. Thus, we do believe that our samples are pure, and we are therefore asking you and reviewer’s indulgence not to add the PXRD pattern in the ‘’Supplementary materials’’ section.
We thank Reviewer 1 for her/his time to study our ms and for the valuable comments provided.
Reviewer 2 Report
The Ms submitted by Dr. Perlepes reports the synthesis and structural characterization of an interesting compound of Zn(II) complex, [ZnCl2(L)] (1) with L =di(2-pyridyl)furoxan [3,4-di(2-pyridyl)-1,2,5-oxadiazole-2-oxide] ligand. The formation of the ligand looks like was obtained through some kind of rearrangement which is promoted by Zn (II), though the reaction of Zn(NO3)2∙6H2O and pyridine-2-chloroxime (ClpaoH) and Et3N reaction mixture in MeOH. Also, the same complex was synthesized through the pre-synthesized ligand L with ZnCl2 in MeOH. The formation of L is completely unpredictable. The compound was thoroughly characterized by NMR and X-ray single crystal structure, and further by DFT calculations and Hirshfeld surface analysis. The work was done in a very satisfactory way. I may suggest some points, which should be taken into consideration:
1) Although the introduction is very informative, it is very long and should be cut by at least 25%.
2) Self citations: There are 116 references in this Ms and at least about 35 references were “self cited”. Some could be related and others are not relevant to this work except it was in the “oxime area”. I think the number of references in general is too much, and should be reduced.
3) I do not think that method (a) for the synthesis of the complex should be considered, in my opinion it should be deleted but it can be addressed in the discussion as this was the one which opened avenues for this work through the other two methods b and c; there was no need for the Eu(NO3)3.
4) Minor typos and errors: lines 97, 189, 207 (during writing this paper), 212 (Studies were ..), 567 (samples are the same or identical, not superimposable).
Overall, this is a nice paper and it deserves the publication as I consider the indicated points are minor issues.

Author Response
1) Although the introduction is very informative, it is very long and should be cut by at least 25%.
The comment is correct. We have drastically condensed ‘’Introduction’’ as recommended.
2) Self citations: There are 116 references in this Ms and at least about 35 references were “self cited”. Some could be related and others are not relevant to this work except it was in the “oxime area”. I think the number of references in general is too much, and should be reduced.
We agree. The number of ‘’self-citations’’ was large. We have deleted 20 self-citations, and now the number of references is 97 (instead of 116 in the initially submitted ms).
3) I do not think that method (a) for the synthesis of the complex should be considered, in my opinion it should be deleted but it can be addressed in the discussion as this was the one which opened avenues for this work through the other two methods b and c; there was no need for the Eu(NO3)3.
We fully agree with the comment. We have deleted method (a) from part 3.3 and this is clearly addressed in the discussion (part 2.1).
4) Minor typos and errors: lines 97, 189, 207 (during writing this paper), 212 (Studies were ..), 567 (samples are the same or identical, not superimposable).
The minor typos and errors have all been corrected in the revised ms.
We thank Reviewer 2 for his/her time to study our ms and the valuable comments provided.
Reviewer 3 Report
In this work the authors report on the reactivity of coordinated chloroximes. They synthesize (using several routes), and spectroscopic characterize a new Zn(II) coordination complex, i.e [ZnCl2(L)] (1); where L is di(2-pyridyl)furoxan [3,4-di(2-pyridyl)-1,2,5-oxadiazole-2-oxide]. The Hirshfeld Surface analysis of the crystal structure performed by the authors allow them to revealing a multitude of intermolecular interactions which generate the final 3D architecture. The complex has also been characterized by the authors using FT IR and Raman spectroscopies. The manuscript also includes the energetic profiles of possible pathways involved in the reaction of ClpaoH and Zn(NO3)2∙6H2O in MeOH in the presence of Et3N.
My opinion is that this manuscript is interesting, it has been competently done and it is technically correct. The topic is adequate for the readership of this journal. The conclusions are well supported by the results. Therefore, I recommend publication after some revision as follows:
1) I suggest the authors including the most relevant noncovalent distances in Figures 5 and 6.
2) I recommend moving Figure 7 and 8 (Raman and NMR spectra) to the ESI.
3) Line 420: DH should be ΔH
4) Ii the energetic profiles (see for instance Figure 10) the authors used ΔH instead of ΔG, is there a reason for this?
5) Line 520: is 1999 correct? I think the Gaussian keyword for this functional is PBE1PBE, please check. Why have the authors selected the PBE0 functional?
6) Is there a reason for not using the def2-tzvp basis set for all atoms?
7) Optional: I would suggest eliminating the first sentence of section 4 (line 601)
Author Response
1) I suggest the authors including the most relevant noncovalent distances in Figures 5 and 6.
The comment is correct and the suggestion valuable. In the previous version of Figure 6b, we had not included Cl⋯π interactions (which were present in Figure 5a and 6a) in order to emphasize the other types of interactions. In the present revised version of both figures, following suggestion by the reviewer, we have removed hydrogen atoms not involved in hydrogen bonds and we have also made thicker the dashed lines representing these interactions. Thus, the intermolecular interactions are clearer and more evident in the revised ms.
2) I recommend moving Figure 7 and 8 (Raman and NMR spectra) to the ESI.
The recommendation seems logical to us. Thus, we have moved Figures 7 and 8 of the initially submitted ms to the ‘’Supplementary materials’’ section. The numbering scheme of the figures in the revised main ms and ‘’Supplementary materials’’ section have been changed accordingly.
3) Line 420: DH should be ΔH
We have corrected the mistake.
4) Ii the energetic profiles (see for instance Figure 10) the authors used ΔH instead of ΔG, is there a reason for this?
There was no special reason for this. The ΔG values are almost the same with the ΔΗ ones, and the results and conclusions concerning thermodynamics would remain practically the same. However, in order to express our respect to the reviewer’s comment, we have included the respective ΔG values in the text. We have also slightly modified the old Figure 10 (Figure 8 in the revised ms) inserting the ΔG values of the various pathways in brackets.
5) Line 520: is 1999 correct? I think the Gaussian keyword for this functional is PBE1PBE, please check. Why have the authors selected the PBE0 functional?
It is correct and, yes, the keyword is PBE1PBE. Concerning the selection of the PBE0 functional, a new sentence has been added in the ‘’Computational Details’’ part 3.5 justifying the selection of this functional.
6) Is there a reason for not using the def2-tzvp basis set for all atoms?
This was done in order to reduce the computational cost.
7) Optional: I would suggest eliminating the first sentence of section 4 (line 601)
Although optional, we have followed the reviewer’s suggestion and eliminated the first sentence of section 4.
We are grateful to the reviewer for his/her time to study our ms and for the valuable comments provided.